Effect of various concentrations of common organic solvents on the growth and proliferation ability of Candida glabrata and their permissible limits for addition in drug susceptibility testing

Liu Juan 1
Zhang Hongxin 2
Zhang Lifang 1
Li Ting 1
Liu Na 3 ynlqdx@sina.com
Liu Qing 1 kqliuqing@sina.com
1 Hebei Key Laboratory of Stomatology, Hebei Clinical Research Center for Oral Disease, School and Hospital of Stomatology, Hebei Medical University , Shijiazhuang, Hebei , China
2 Department of Emergency, The Second Hospital of Hebei Medical University , Shijiazhuang, Hebei , China
3 Department of Preventive Dentistry, School and Hospital of Stomatology, Hebei Medical University , Shijiazhuang, Hebei , China
Uversky Vladimir
Electronic publication date: 2023 Nov 21
Publication date: 2023
Volume: 11
Electronic Location ID: e16444
Received 2023 Jul 7; Accepted 2023 Oct 20
Copyright: © 2023 Liu et al.
Copyright year: 2023
Copyright holder: Liu et al.
License: This is an open access article distributed under the terms of the Creative Commons Attribution License, which permits unrestricted use, distribution, reproduction and adaptation in any medium and for any purpose provided that it is properly attributed. For attribution, the original author(s), title, publication source (PeerJ) and either DOI or URL of the article must be cited.
License URL: https://creativecommons.org/licenses/by/4.0/

Keywords: Organic solvent, Candida glabrata, Growth and proliferation ability

Funding: Hebei Provincial Health Commission 20231087 This research was supported by the Medical Science Research Project of Hebei Provincial Health Commission (20231087). The funders had no role in study design, data collection and analysis, decision to publish, or preparation of the manuscript.

==============================
Objectives

Dimethyl sulfoxide (DMSO), acetone, ethanol, and methanol are organic solvents commonly used for dissolving drugs in antimicrobial susceptibility testing. However, these solvents have certain antimicrobial activity. Currently, standardized criteria for the selection and dosage of drug solvents in drug susceptibility testing research are lacking. The study aims to provide experimental evidence for the selection and addition limit of drug solvents for the in vitro antifungal susceptibility test of Candida glabrata (C. glabrata).

Methods

According to the recommendation of the Clinical and Laboratory Standards Institute (CLSI) M27-A3, a 0.5 McFarland C. glabrata suspension was prepared and then diluted 1:1,000. Next, a gradient dilution method was used to prepare 20%, 10%, 5%, and 2.5% DMSO/acetone/ethanol/methanol. The mixture was plated onto a 96-well plate and incubated at a constant temperature of 35 °C for 48 h. The inhibitory effects of DMSO, acetone, ethanol, and methanol on C. glabrata growth and proliferation were analyzed by measuring optical density values at 600 nm (OD600 values).

Results

After 48 h incubation, the OD600 values of C. glabrata decreased to different extents in the presence of the four common organic solvents. The decrease in the OD600 values was greater with increasing concentrations within the experimental concentration range. When DMSO and acetone concentrations were higher than 2.5% (containing 2.5%) and methanol and ethanol concentrations were higher than 5.0% (containing 5.0%), the differences were statistically significant compared with the growth control wells without any organic solvent (P < 0.05).

Conclusion

All four organic solvents could inhibit C. glabrata growth and proliferation. When used as solvents for drug sensitivity testing in C. glabrata, the concentrations of DMSO, acetone, ethanol, and methanol should be below 2.5%, 2.5%, 5%, and 5%, respectively.

Introduction

Being integral to the normal microbiota, Candida is commonly found in the human body and environment. However, under compromised host immunity or dysbiosis, Candida can use virulence factors and actively contribute to the pathophysiological development of disease and initiate host infections (Silva et al., 2012).

Candida albicans (C. albicans) is the predominant fungal pathogen in epidemiological studies. However, reports on drug resistance in C. albicans are increasing, in addition to the detection rate of non-C. albicans (NCA) species. After C. albicans, C. glabrata is the second most prevalent pathogen in candidiasis (Hassan, Chew & Than, 2021). C. glabrata is frequently detected in the oral lesions of HIV-infected individuals (8.8%) and within peri-implant pockets of patients with implant-associated infections (14.2%), second only to C. albicans (Khedri et al., 2018; Yang et al., 2015). Epidemiological data for the past two decades demonstrates a decline in C. albicans prevalence among patients with vulvovaginal candidiasis (VVC), while the prevalence of NCA, particularly C. glabrata, has exhibited an upward trend, thereby becoming the most common pathogen (Narayankhedkar, Hodiwala & Mane, 2015; Wang et al., 2016). C. glabrata accounts for approximately one-half to two-thirds of NCA-associated VVC cases (Makanjuola, Bongomin & Fayemiwo, 2018). Notably, C. glabrata-induced invasive infections are associated with a higher mortality rate than other NCA species, with an estimated fatality rate nearing 50% (Eliaš & Gbelská, 2022; Nishikawa et al., 2016).

Because of the increasing drug resistance in C. albicans and the rising detection rate of NCA, new antifungal compounds are being continuously searched. Encouragingly, natural compounds such as Eugenia uniflora extract (Souza et al., 2018), Phyllanthus emblica Linn. (Thaweboon & Thaweboon, 2011), eugenol (de Paula et al., 2014), and phlorotannins (Lopes et al., 2013) have exhibited significant potential in the field of antifungal research. However, solvents with distinct polarities, commonly including methanol, ethanol, and acetone, are required for extracting these antifungal compounds.

In vitro antifungal susceptibility testing, as a method for determining the inhibitory activity of drugs against pathogenic fungi such as C. albicans, Candida tropicalis (C. tropicalis), C. glabrata, and Cryptococcus neoformans, is a crucial role in fungal drug resistance surveillance, epidemiological research, and comparative evaluation of in vitro antifungal activities of existing drugs (Subcommittee on Antifungal Susceptibility Testing of the ESCMID European Committee for Antimicrobial Susceptibility Testing, 2008). Microdilution is commonly used in antifungal susceptibility testing for its advantages of minimal sample usage, ease of control, standardized operation, and simultaneous acquisition of quantitative and qualitative results (Cai et al., 2011; Jorgensen & Ferraro, 2009). However, the test compound needs to be diluted multiple times in a suitable solvent for this technique. Therefore, organic solvents, such as methanol, ethanol, acetone, and Dimethyl sulfoxide (DMSO), are frequently used in drug sensitivity tests (Dyrda et al., 2019; Wadhwani et al., 2008). Lopes et al. (2017) used DMSO as a solvent while studying the inhibitory effects of flavonoids on Staphylococcus aureus. Methanol was used as the drug solvent when Gaucher et al. (2013) investigated the antibacterial effects of resveratrol. Different solvent types and concentrations can have varying effects on microbial growth (Dyrda et al., 2019; Wadhwani et al., 2008; Kirkwood et al., 2018). Thus, the influence of organic solvents on microorganisms in susceptibility testing and the potential variability in sensitivity among different microbes must be considered. To date, the effects of various solvents on C. albicans growth has been extensively examined, as evidenced in studies by Akram Randhawa (2008), Eloff, Masoko & Picard (2007), Chauhan, Raut & Karuppayil (2011), Rane et al. (2012), and Peters et al. (2013). However, studies specifically addressing the most prevalent NCA, particularly C. glabrata, which has the highest detection rate, are scarce. Knowledge regarding the influence of distinct concentrations of common organic solvents (e.g., methanol, ethanol, DMSO, and acetone) on C. glabrata growth during susceptibility testing is limited. Consequently, our study focuses on C. glabrata to address this gap in understanding.

Some of the methods commonly used for assessing fungal growth are optical density measurement (OD value), cell counting, and dry weight determination. de-Souza-Silva et al. (2018) reported the use of OD600 for evaluating antifungal susceptibility in drug sensitivity testing. Compared with other methods, the OD600 measurement is relatively fast, convenient, and easy to perform. This technique has been widely applied in various studies and experiments (Silva et al., 2022; Radhakrishnan et al., 2018; Gaucher et al., 2013). The OD600 value is a critical measure in drug sensitivity testing that reflects the growth and proliferation ability of Candida, with a positive linear correlation between the shape of the regular Candida suspension and OD600. Higher OD600 values indicate more robust growth (Lei et al., 2014). In our study, we applied the microdilution method and OD600 values to assess the impact of varying concentrations of organic solvents (DMSO, acetone, ethanol, and methanol) on C. glabrata growth and proliferation. Our findings provided reference values for the concentrations of organic solvents to be used in drug sensitivity testing for C. glabrata.

Materials and Methods

Strains

C. glabrata was isolated from the Laboratory Microbiology Room of Bethune International Peace Hospital, identified by using the MA120 Microbial Identification Instrument (Meihua Med Tech, Zhuhai, China) (He et al., 2021), and frozen at −80 °C until use.

Main instruments and reagents

McFarland densitometer (Meihua Med Tech, Zhuhai, China), microplate reader (MDC, the US), thermostatic incubator (Xinmiao, Shanghai, China), autoclave (Xinhua Medical, Shandong, China), biosafety cabinet (Heal Force, Hong Kong, China), DMSO (GR; Solarbio, Beijing, China), acetone (GR; Innochem, Beijing, China), methanol (GR; Chuangshi, Jinan, China), ethanol (GR; Titan, Shanghai, China), RPMI1640 medium (Gibco; Thermo Fisher Scientific Technology Co., Ltd., Waltham, MA, USA), Sterile saline solution (NaCl 0.85 g/100 mL) (NO. 4 Pharmaceutical, Shijiazhuang, China), and CHROMager and Sabouraud Dextrose Agar (Shanghai Comagal Microbial Technology Co., Ltd., Shanghai, China) were used in this study.

Methods

Organic solvent preparation

The gradient dilution method was used to prepare varied organic solvent concentrations. Four centrifuge tubes (No. 1 to No. 4) were employed as per the protocol. Among the four tubes, 5 mL of RPMI 1640 culture medium was added to each of tubes No. 2 to No. 4. Then, 2 mL DMSO/acetone/ethanol/methanol and 8 mL RPMI 1640 culture medium were added to tube No. 1. Subsequently, 5 mL of the mixture from tube No. 1 was transferred to tube No. 2 and thoroughly mixed. The same steps were sequentially repeated until tube No. 4 was reached. Consequently, DMSO/acetone/ethanol/methanol solutions of 20%, 10%, 5%, and 2.5% concentrations (v/v%) were successfully obtained.

Effects of four common organic solvents on C. glabrata growth and proliferation

According to the CLSI M27-A3 standard, five single colonies (diameter: approximately 1 mm), which had grown on Sabouraud dextrose agar (SDA) for 24 h, were selected (Fig. 1). The colonies were suspended in 5 mL of sterile saline solution (NaCl 0.85 g/100 mL). The concentration was adjusted to 0.5 McFarland (≈1–5 × 106 CFU/mL), and then, the C. glabrata suspension was diluted with RPMI 1640 liquid medium containing 165 mM of 3-Morpholinopropanesulfoinc acid (MOPS) at 1:1,000 (first 1:20, then 1:50). Finally, a 2× C. glabrata working suspension was obtained (≈1–5 × 103 CFU/mL).

Figure 1 Candida glabrata single colony.

Then, 100 uL/well of 2×DMSO/acetone/ethanol/methanol with 20%, 10%, 5%, and 2.5% concentrations were added to the 96-well plate in advance, followed by the addition of 100 uL/well of 2× C. glabrata working suspension to achieve a final concentration of DMSO/acetone/ethanol/methanol 10%, 5%, 2.5%, and 1.25%. A growth control well (100 μL RPMI 1640 liquid medium +100 μL 2× C. glabrata working suspension) without DMSO/acetone/ethanol/methanol and a blank control well containing only RPMI 1640 liquid medium (200 μL RPMI 1640 liquid medium) were maintained, and 200 μL phosphate buffer solution (PBS) was added to the outermost circle of the 96-well plate so as to prevent the culture solution from evaporating, after which the plate was cultured at 35 °C for 48 h. The effect of organic solvents on C. glabrata growth and proliferation is expressed by the growth inhibition rate I (%):

I(%)=A1−A2A1−A0×100%

I (%) represents the rate of C. glabrata growth inhibition by the organic solvent, A0 represents the OD600 value of the blank control well, A1 represents the OD600 value of the growth control well, and A2 represents the OD600 value after treatment with varying concentrations of organic solvents.

Statistical analysis

SPSS 27.0 software (IBM Corp., Armonk, NY, USA) was used for data analysis. The data were first examined for normality. If the data were normally distributed, mean ± standard deviation ( x¯ ± s) was used for statistical description and ANOVA was used for comparison. If the data showed no normal distribution, median (M) and quartiles (P25, P75) were used for statistical description and the Kruskal–Wallis H rank sum test for analysis.

Results

Table 1 presents OD600 values of C. glabrata in the presence of different concentrations of various organic solvents. The OD600 values of C. glabrata decreased to varying extents after exposure to the four organic solvents for 48 h, The decrease occurred in a concentration-dependent manner within the experimental concentration range. This means the OD600 value was decreased with the increase in the concentration of the organic solvent, which indicated that C. glabrata growth and proliferation were significantly inhibited.

Table 1 Effect of different concentrations of common organic solvents on optical density values (OD600) of Candida glabrata.

Solvent	Concentration (v/v%)	Mean	Percentile	
P 25	P 50	P 75	
DMSO	10	0.045	0.044	0.046	0.047	
5	0.864	0.856	0.864	0.874	
2.5	1.029	1.013	1.033	1.043	
1.25	1.079	1.071	1.079	1.088	
0	1.136	1.133	1.137	1.139	
Acetone	10	0.398	0.347	0.392	0.444	
5	1.080	1.072	1.075	1.109	
2.5	1. 071	1.051	1.100	1.106	
1.25	1.102	1.092	1.108	1.111	
0	1.132	1.122	1.131	1.140	
Ethanol	10	0.877	0.835	0.853	0.939	
5	1.052	1.015	1.056	1.086	
2.5	1.221	1.209	1.224	1.234	
1.25	1.236	1.228	1.238	1.240	
0	1.250	1.244	1.251	1.253	
Methanol	10	1.122	1.116	1.125	1.137	
5	1.177	1.169	1.177	1.187	
2.5	1.191	1.182	1.189	1.200	
1.25	1.200	1.194	1.203	1.208	
0	1.212	1.203	1.208	1.222	

Figure 2 presents the growth inhibition rates of various organic solvents at differing concentrations on C. glabrata. As the concentration increased, the growth inhibition rates of the four organic solvents against C. glabrata significantly increased. The growth inhibition rates of 10% DMSO, 10% acetone, and 10% ethanol were 100%, 67.90% and 32.95%, respectively. The growth inhibition rate of 5% DMSO and 5% ethanol against C. glabrata were 24.93% and 16.4%, respectively. The growth inhibition rates were <10% when the DMSO and ethanol concentrations were <2.5% (containing 2.5%), the acetone concentration was <5% (containing 5%), and the methanol concentration was <10% (containing 10%).

Figure 2 Inhibitory rate of organic solvent with different concentrations on growth of Candida glabrata.

Discussion

Dissolving most common antifungal drugs and natural compounds in water is difficult because they are mostly macromolecular compounds. Therefore, methanol, ethanol, Tween 80, and DMSO are often used as drug solvents. Because of the different drug types, physical and chemical properties, and microbial sensitivity to solvents, the solvent types and concentrations used in the drug sensitivity test also differ, lacking a unified specification (Lee & Kim, 2022; Demolsky, Sugiaman & Pranata, 2022). We here investigated the effects of DMSO, acetone, ethanol, and methanol on the growth and proliferation of C. glabrata, the most frequently detected NCA species. The study aimed to provide experimental evidence for the selection and limited addition of organic solvents and their concentrations in antifungal susceptibility testing.

Owing to the variations in the solubility of natural compounds, poorly soluble compounds may require higher concentrations of organic solvents for dissolution. To ensure the tested antifungal compounds are adequately dissolved in the organic solvents used in the antifungal susceptibility testing, and to determine and consider the potential impact of solvents themselves on C. glabrata, multiple concentration ranges may need to be tested to minimize or avoid the solvent’s influence on the fungus and obtain reliable experimental results. Therefore, we selected four concentrations for discussion: lower concentrations of 1.25% and 2.5%, an intermediate concentration of 5%, and a higher concentration of 10%.

DMSO is an aprotic solvent miscible with water and other organic solvents. It is often used as a cell permeability enhancer, cryoprotectant, transdermal agent, anti-inflammatory agent, and solvent for drugs dissolution (Modrzyński, Christensen & Brandt, 2019). However, the inhibitory effect of DMSO on Candida growth has also been frequently reported. For example, Akram Randhawa (2008) found that DMSO could affect C. albicans growth and germ tube germination. The growth of germ tubes was inversely proportional to the DMSO concentration. At 10% DMSO, the germination of C. albicans germ tubes was completely inhibited. In this experiment, the effect of different concentrations of DMSO (1.25–10%) on C. glabrata growth was studied. The OD600 value of C. glabrata was significantly lower in the presence of 2.5% DMSO than in the growth control well without DMSO (P < 0.05), indicating that 2.5% DMSO significantly inhibited C. glabrata growth (Table 2). This is similar to the result of Rodríguez-Tudela et al. (2001) who reported that 2% DMSO significantly inhibited the growth of C. glabrata and C. albicans. The OD600 value of C. glabrata in the presence of 10% DMSO significantly reduced compared with the growth control well without DMSO. C. glabrata growth was significantly inhibited (P < 0.01), which is consistent with the significant inhibition effect of 10% DMSO on C. albicans reported by Demolsky, Sugiaman & Pranata (2022). However, Hazen (2013) reported a complex effect of low DMSO concentration (<4%) on C. albicans growth, which might inhibit, promote, or have no effect on C. albicans growth. In the present study, no such complex phenomenon of the effect of DMSO on C. glabrata was observed, which might be related to the unique physicochemical properties of different strains and their varied sensitivities to DMSO.

Table 2 Effect of different concentrations of DMSO on the optical density values (OD600) of Candida glabrata.

Concentration of DMSO (v/v%)	M (P25, P75)	Kruskal-Wallis H rank sum test	
H value	P value	
10**	0.046 (0.044, 0.047)	42.294	<0.001	
5**	0.864 (0.856, 0.874)	
2.5*	1.033 (1.013, 1.043)	
1.25	1.079 (1.071, 1.088)	
0	1.137 (1.133, 1.139)	
Notes:

* Compared with 0%DMSO, P < 0.05.

** Compared with 0%DMSO, P < 0.01.

H value, The test statistic for the Kruskal-Wallis H rank sum test.

M(P25, P75), median and quartiles.

Acetone could inhibit the growth of C. albicans and Cryptococcus neoformans at a minimum inhibitory concentration (MIC, the lowest concentration that completely inhibits the growth of a microorganism) of 48% and 56%, respectively (Eloff, Masoko & Picard, 2007). However, the effect of acetone on C. glabrata growth is very unclear. We here explored the effect of 1.25–10% acetone for 48 h on C. glabrata growth. Compared with the growth control well, a significant decrease in the OD600 value of C. glabrata was observed at 2.5% (P < 0.05), with a growth inhibition rate of 2.94%. By contrast, 10% acetone resulted in a growth inhibition rate of 67.9% (Table 3 and Fig. 2). Acetone can lead to a reduction in nuclear contents and damage the plasma and nuclear membrane integrity (Hoetelmans et al., 2001), which may serve as a reference value for studying the inhibitory mechanism of acetone against C. glabrata growth. However, the specific mechanism needs to be further explored. Acetone was also toxic to Salmonella typhimurium and Escherichia coli, and 7.7% acetone (with cell viability of 78–95%) was more toxic to the experimental strain than 4.0% acetone (with cell viability of 83–100%) (Shibata et al., 2020). This suggested that the toxicity of acetone to microbes increases with an increase in concentration, which is consistent with our study results on C. glabrata.

Table 3 Effect of different concentrations of acetone on the optical density values (OD600) of Candida glabrata.

Concentration of acetone (v/v%)	M (P25, P75)	Kruskal-Wallis H rank sum test	
H value	P value	
10**	0.392 (0.347, 0.444)	36.293	<0.001	
5*	1.075 (1.072, 1.109)	
2.5*	1.100 (1.051, 1.106)	
1.25	1.108 (1.092, 1.111)	
0	1.131 (1.122, 1.140)	
Notes:

* Compared with 0% acetone, P < 0.05.

** Compared with 0% acetone, P < 0.01.

H value, The test statistic for the Kruskal-Wallis H rank sum test.

M(P25, P75), median and quartiles.

Ethanol is an inhibitor of bacterial and fungi growth. It can inhibit the formation of mycotoxin aflatoxin B1 (Ma et al., 2019; Ren et al., 2020). In microbes, ethanol can denature the protein, destroy the cell wall and the enzyme system, and interfere with metabolism, thereby achieving the bactericidal effect (Hamida, 2019). As a disinfectant, the commonly used ethanol concentration is 75%. Ethanol solutions at a relatively low concentration can still kill bacteria (Hamida, 2019; Sauerbrei, 2020). Therefore, the selection and applicability of the ethanol concentration in experiments is extremely crucial, especially in the related research on Candida. Varying concentrations of the ethanol solvent can have different degrees of effect on C. albicans, which is currently the most thoroughly studied organism. Ethanol at 4% concentration not only inhibits germ tube formation and length in C. albicans but also significantly affects the organism’s growth and viability as well as biofilm formation (Chauhan, Raut & Karuppayil, 2011). The ethanol concentration of 10% or even higher reduces the formation of C. albicans biofilm by >99% (Rane et al., 2012). An ethanol concentration of ≥20% can kill biofilm and of ≥30% can completely inhibit the metabolic activity when it acts on the C. albicans biofilm for 4 h (Peters et al., 2013). However, related research on the effect of ethanol on C. glabrata is relatively rare. We explored whether the ethanol solvent can inhibit C. glabrata growth as well as the trend of inhibition at different ethanol concentrations. Ethanol at a relatively low concentration (≤10%) inhibited C. glabrata. The higher the ethanol concentration, the more obvious the inhibition. At 5% ethanol, the OD600 value of C. glabrata significantly decreased compared with the growth control well (P < 0.01), and the inhibition was significant, with the inhibition rate of 15.56%, The inhibition rate of 32.95% was observed at the highest concentration (10%) used in the experiment (Table 4 and Fig. 2).

Table 4 Effect of different concentrations of ethanol on the optical density values (OD600) of Candida glabrata.

Concentration of ethanol (v/v%)	M (P25, P75)	Kruskal-Wallis H rank sum test	
H value	P value	
10**	0.853 (0.835, 0.939)	40.650	<0.001	
5**	1.056 (1.015, 1.086)	
2.5	1.224 (1.209, 1.234)	
1.25	1.238 (1.228, 1.240)	
0	1.251 (1.244, 1.253)	
Notes:

** Compared with 0% ethanol, P < 0.01.

H value, The test statistic for the Kruskal-Wallis H rank sum test.

M(P25, P75), median and quartiles.

Hamida (2019) reported that methanol had a certain bactericidal effect on Staphylococcus, and the higher the methanol concentration, the longer the exposure time, and the better the bactericidal effect. According to Matzneller, Manafi & Zeitlinger (2011), the MIC value of statins in the presence of the 5% methanol solvent was >10 times higher than that in the presence of the 100% methanol solvent. This indicated that the antimicrobial effect of statins depended on the presence of the organic solvent methanol. On the one hand, these studies have shown that methanol exerts different degrees of toxicity to microbes, and the magnitude of toxicity is closely related to the methanol concentration and the types of microbes. On the other hand, in the drug sensitivity test, the effect of methanol as a solvent on the microbes could not be ignored. We here found that methanol at the experimental concentrations of 10%, 5%, 2.5%, and 1.25% inhibited C. glabrata growth, and the growth inhibition rate was positively correlated with the methanol concentration. The concentration of <5% (including 5%) had a weak inhibitory effect on C. glabrata, with the highest inhibition rate being 2.50%. At 10% methanol, the OD600 value of C. glabrata decreased significantly (P < 0.01), the inhibition rate was 7.73%, and the inhibition was significant (Table 5 and Fig. 2). As the concentration in our study was limited, the effect of higher methanol concentration on C. glabrata was unknown.

Table 5 Effect of different concentrations of methanol on the optical density values (OD600) of Candida glabrata.

Concentration of methanol (v/v%)	M (P25, P75)	Kruskal-Wallis H rank sum test	
H value	P value	
10**	1.125 (1.116, 1.137)	35.315	<0.001	
5**	1.177 (1.169, 1.187)	
2.5	1.189 (1.182, 1.200)	
1.25	1.203 (1.194, 1.208)	
0	1.208 (1.203, 1.222)	
Notes:

** Compared with 0% methanol, P < 0.01.

H value, The test statistic for the Kruskal-Wallis H rank sum test.

M (P25, P75), median and quartiles.

Furthermore, among the four solvents within the same concentration group, methanol displayed the lowest inhibition rate against C. glabrata. Studies have reported low inhibition rates of methanol against bacteria. Dyrda et al. (2019) found that among 4.8% concentrations of methanol, ethanol, acetone, N,N dimethylformamide, DMSO, and nujol, the methanol group had the highest number of viable Bacillus subtilis cells. However, another study investigated the effects of various organic solvents on S. epidermidis MTCC 435, Pseudomonas oleovorans MTCC 617, Vibrio cholerae MTCC 3906, Shigella flexneri MTCC 1457, and Salmonella paratyphi A (Wadhwani et al., 2008). In that study, methanol was more toxic than DMSO at concentrations of 1–3%, whereas the reverse trend was observed within 4–6%. This indicates that the inhibitory effect of methanol depends on its concentration and microbial species, and each has its specific mechanism. As suggested by Weber & de Bont (1996), the response of the microbe’s cellular system under solvent stress is basically a result of changes both in the membrane lipid composition and in the protein, sterol, hopanoid, and carotenoid content, which modify the plasma membrane properties (e.g., fluidity, membrane permeability, and rigidity). However, the action mechanism of methanol on certain fungi has not been completely elucidated, especially on C. glabrata. We have been unable to further explore the mechanism through which the methanol solvent has a slight effect on C. glabrata in our experiment. This may be related to both the chemical and physical effects of methanol. On one hand, C. glabrata may be less sensitive to methanol. The distribution, transport, and chemical reactions of methanol in the cell membrane or cytoplasm may differ compared with those of other organic solvents. On the other hand, methanol may alter cell membrane properties, including fluidity, membrane permeability, and rigidity, thus improving membrane resistance and reducing the amount of methanol entering the cell.

Taken together, DMSO, acetone, ethanol, and methanol have inhibitory effects on various microorganisms. Concentration selection is a key factor that cannot be ignored when using these compounds as drug solvents for drug sensitivity tests. In this study, the microdilution method was used to study the effects of DMSO, acetone, ethanol, and methanol on C. glabrata growth. Different solvents had varying inhibition degrees against C. glabrata. For example, when the concentration of organic solvents was 1.25%, 2.5% and 10%, the inhibition rates of DMSO and acetone were higher than those of methanol and ethanol. At 5% concentration, the inhibition rates of DMSO and ethanol against C. glabrata were higher than those of acetone and methanol. However, the inhibition rates of all four solvents against C. glabrata followed the concentration effect. This effect indicated that the influence of DMSO, acetone, ethanol, and methanol on C. glabrata was related to the solvent type and concentration. This suggests that the organic solvent types used in the drug sensitivity test in vitro should be carefully selected according to the actual needs, The concentration of organic solvents should be reduced as much as possible to reduce its influence on the tested microorganisms.

In this study, the differences between the growth control wells and the wells containing 2.5% DMSO, 2.5% acetone, 5% ethanol, and 5% methanol were statistically significant. Therefore, when conducting drug susceptibility tests on C. glabrata, the DMSO, acetone, ethanol, and methanol concentrations should be below 2.5%, 2.5%, 5%, and 5%, respectively. However, due to limitations in our experimental design, further research is warranted to investigate the effects of a wider concentration range of organic solvents on C. glabrata growth and proliferation.

Conclusion

In summary, all four organic solvents inhibited C. glabrata growth and proliferation, with a more pronounced inhibition observed as the concentration increased within the experimental range. Among these solvents at the same concentration, DMSO, acetone, and ethanol exhibited a more significant inhibitory effect on C. glabrata growth and proliferation compared with methanol. To avoid the potential impact of the solvents themselves on C. glabrata during drug susceptibility testing, the DMSO, acetone, ethanol, and methanol concentrations should be maintained below 2.5%, 2.5%, 5%, and 5%, respectively.

Supplemental Information

Supplemental Information 1 Raw data.

Click here for additional data file.

Additional Information and Declarations

Competing Interests

Author Contributions

Data Availability

The authors declare that they have no competing interests.

Juan Liu conceived and designed the experiments, performed the experiments, analyzed the data, prepared figures and/or tables, and approved the final draft.

Hongxin Zhang performed the experiments, analyzed the data, prepared figures and/or tables, and approved the final draft.

Lifang Zhang performed the experiments, analyzed the data, prepared figures and/or tables, and approved the final draft.

Ting Li performed the experiments, analyzed the data, prepared figures and/or tables, and approved the final draft.

Na Liu conceived and designed the experiments, authored or reviewed drafts of the article, and approved the final draft.

Qing Liu conceived and designed the experiments, authored or reviewed drafts of the article, and approved the final draft.

The following information was supplied regarding data availability:

The raw data are available in the Supplemental File.

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
