# Peer review of "Effect of various concentrations of common organic solvents on the growth and proliferation ability of Candida glabrata and their permissible limits for addition in drug susceptibility testing"

_PeerJ, doi:10.7717/peerj.16444_

## Round 0.1 · original submission · Major Revisions

As you can see, all three reviewers raised some serious concerns about your study and requested major revisions. Please address all issues indicated by the reviewers and amend your manuscript accordingly.

**Language Note:** The review process has identified that the English language must be improved. PeerJ can provide language editing services - please contact us at [email protected] for pricing (be sure to provide your manuscript number and title). Alternatively, you should make your own arrangements to improve the language quality and provide details in your response letter. – PeerJ Staff

Reviewer 1 ·

Basic reporting

In general, the research work is of interest due to the relevance of C. glabrata in the clinical setting and the antifungals that can be used to inhibit its growth.

Experimental design

The design of the experiments is adequate, in addition, the methodologies used to evaluate the growth or inhibition of a microorganism have been used.
However, I consider that the methodology described in the manuscript needs to be restructured or improved, so that it is not confusing. In addition, there are some considerations that must be taken into account in order to accept the conditions or parameters used in the experiments. I'll detail them shortly.

Validity of the findings

The results were as expected. Because the growth of the organisms in the evaluated solvents would have to be affected. However, I believe that the reason why these concentrations were chosen should be mentioned. Are these concentrations used to dissolve some antifungal compounds? or that the experiments be carried out with a greater range of concentrations.

Additional comments

Some considerations:

The introduction section could be improved, it could mention those compounds most used to combat Candida and those solvents that have been used to dissolve said antifungal agents.
You could also mention the effect of solvents on cells.
Mention why C. glabrata has been selected; Since there are other species, why this one in particular?

Line 34. Mention of some pathogenic fungi.
Line 54-56: Reference of this identification.
Lines 67-72: Reference required. The description of method is confusing .
Lines 76-80: Define SDA
Sterile saline solution? what solution?
Define Mc
Line 85: without any solvent.
Line 87: define PBS
Line 88: Why was it only measured at this time? Is it not relevant at 24 or 72 h?
Why was it carried out at 35C and not at 28C or 37C?

28 C Normally grows best and
37 C The temperature of the host

Line 133: Full name of C. albicans

Line 139-146: Consider reviewing these lines, it is too much difference between 2.5 and 10%. What is more sensitive to 2.5 than to 10%? Perhaps there is an error in these data.

Line 153: Define MIC

Line: 185: Is there any hypothesis as to why this solvent has little inhibition?


In general, in the manuscript it is referred to as the activity of C. glabrata. What activities are you referring to?

I believe that it could only refer to the proliferation capacity of the fungi.

Reviewer 2 ·

Basic reporting

Summary
In this manuscript, the author investigates the effect of four different organic solvents on the growth activity of C. glabrata. It was discovered through the microdilution method that all four organic solvents (DMSO, acetone, methanol, and ethanol) have an inhibitory effect on the growth of said fungus. It is reported that at a concentration equal to or greater than 2.5%, DMSO and acetone weaken the proliferation capacity and activity of C. glabrata while methanol and ethanol require a minimum concentration of 5% to achieve such effect. The inhibitory effect is validated by a statistically significant difference between the test groups and a control group with no organic solvent treatment.

Basic Reporting
The English in the article needs to be revised in order to not cause any confusion. Parts of the logic in the introduction section are hard to follow and in the methods section, the experimental setup is somewhat confusing.
The novelty is questionable, as organics solvents such as methanol and ethanol are very commonly used for anti-bacterial, anti-viral, and anti-fungal purposes. There have been studies on the effect of DMSO on C. glabrata growth. More evidence is needed to justify the novelty and significance of this study. I do not see an unsolved problem being raised by the author. It is recommended that the introduction part be worked on and improved to a great degree. Specifically, how is this study different from the previous studies on the effects of organic solvents on C. glabrata? What kind of unique question is addressed by this article?
Problems with experimental setup and results/discussions will follow in subsequent sections.

Experimental design

Experimental Design
1. Line 67-72. This part is extremely confusing. What is added and in what order? Was RPMI added twice before and after organic solvents in tube no.1? Was it a typo or was if done on purpose? If so, why?
2. Line 74-80. How isolated are the single colonies? It would be nice to have pictures.
3. There needs to be more statistical analysis. For example, utilizing ANOVA, to determine the difference between growth control group and 1.25% organic solvent groups is statistically significant.

Validity of the findings

Validity of Findings
1. Please also provide a normalized result of OD. The absolute number might mean something when comparing between different experimental groups but when it comes to growth, normalized data is more convincing.
2. Are conclusions based on the 10 replicates provided in the raw data table? If so, please update error bar in Figure 1 as well. Otherwise, the figure lacks statistical significance.
3. Can you also provide growth curve in all experiments? You are addressing the problem of growth yet only an endpoint is provided. With growth data, the results might be more convincing and more information can be extracted from the experiments.

Additional comments

Please translate the raw data table into English.

Reviewer 3 ·

Basic reporting

I strongly recommend using a scientific English editing service to address the inconsistencies in the writing style and to ensure the content is presented in an impactful manner. The current manuscript is weak, but thorough reading, writing, and careful data interpretation can lead to substantial improvement.

Experimental design

The literature references provided are generally sufficient, and the figure and tables are presented in an acceptable manner. The introduction needs more description to establish the validity of the analytical methods used.
The Materials and Methods section is adequately described; however, it would benefit from some language refinement to improve clarity.

Validity of the findings

The manuscript analyses the inhibitory role of 4 solvents in the growth of the fungi Candida glabrata to address the optimal concentration aspects in drug sensitivity tests. Though the data is robust, enhancing discussion and explanation is advised. I have provided some corrections and queries for improving the manuscript.

Annotated reviews are not available for download in order to protect the identity of reviewers who chose to remain anonymous.

---

## Round 0.2 · accepted · Accept

All issues pointed out by the reviewers were addressed and the revised manuscript is acceptable.

Reviewer 3 ·

Basic reporting

No comment

Experimental design

No comment

Validity of the findings

No comment

Additional comments

All my queries were answered satisfactorily. I would recommend the manuscript for acceptance.